# Evaluation of 11 DNA Automated Extraction Protocols for the Detection of the 5 Mains Candida Species from Artificially Spiked Blood

**DOI:** 10.3390/jof7030228

**Published:** 2021-03-19

**Authors:** Estelle Menu, Jordi Landier, Elsa Prudent, Stéphane Ranque, Coralie L’Ollivier

**Affiliations:** 1Institut de Recherche pour le Développement, Aix Marseille Université, Assistance Publique-Hôpitaux de Marseille, Service de Santé des Armées, VITROME: Vecteurs—Infections Tropicales et Méditerranéennes, 13385 Marseille, France; Stephane.ranque@ap-hm.fr (S.R.); Coralie.lollivier@ap-hm.fr (C.L.); 2IHU Méditerranée Infection, 13385 Marseille, France; Elsa.prudent@ap-hm.fr; 3Sciences Economiques & Sociales de la Santé & Traitement de l’Information Médicale, Institut de Recherche pour le Développement, Institut National de Santé et de Recherche Médicale, Aix Marseille Université, 13385 Marseille, France; jordi.landier@ird.fr

**Keywords:** *Candida*, candidaemia, DNA extraction

## Abstract

The molecular detection of *Candida* plays an important role in the diagnosis of candidaemia, a major cause of morbidity and mortality. The sensitivity of this diagnosis is partly related to the efficiency of yeast DNA extraction. In this monocentric study, we investigated the suitability of 11 recent automated procedures for the extraction of low and high amounts of *Candida* DNA from spiked blood. The efficacy of the DNA extraction procedures to detect *Candida* spp. in blood samples ranged from 31.4% to 80.6%. The NucliSENS^TM^ easyMAG^TM^ procedure was the most efficient, for each species and each inoculum. It significantly outperformed the other procedures at the lower *Candida* inocula mimicking the clinical setting. This study highlighted a heterogeneity in DNA extraction efficacy between the five main *Candida* species (*Candida albicans*, *Candida glabrata*, *Candida parapsilosis*, *Candida tropicalis* and *Candida krusei*). Up to five automated procedures were appropriate for *C. krusei* DNA extraction, whereas only one method yielded an appropriate detection of low amount of *C. tropicalis*. In the era of the syndromic approach to bloodstream infection diagnosis, this evaluation of 11 automated DNA extraction methods for the PCR diagnosis of candidaemia, puts the choice of an appropriate method in routine diagnosis within the reach of laboratories.

## 1. Introduction

*Candida* species are among the top five pathogens associated with health-care bloodstream infections [1], carrying a high attributable mortality of up to 40% [2,3,4]. Major risk factors for developing candidaemia have well been identified and include critical illness, long-term stay in an intensive care unit, abdominal surgery, malignant haematologic diseases, intravenous catheter, parenteral nutrition, and administration of broad-spectrum antibacterial therapy [5]. Early diagnosis is critical for appropriate patient management and for improving candidaemia outcomes. Blood cultures, the current diagnostic gold standard, are limited by low sensitivity, ranging from 21% to 71% [6], and a slow turnaround, usually exceeding 48 h [6,7,8].

Recently, several in-house or commercially available polymerase chain reaction (PCR) assay kits have been developed. Generally, these assays target the five main *Candida* species involved in candidaemia: *Candida albicans*, *Candida glabrata*, *Candida parapsilosis*, *Candida tropicalis*, and *Candida krusei* [9,10]. In contrast to blood cultures, PCR allows for the rapid and specific detection of yeasts within whole blood, serum, or plasma samples, without requiring the sampling of large blood volumes and prior cultivation [10]. Interestingly, PCR in blood samples has shown >90% specificity, and up to 100% sensitivity. These diagnostic indices are better than those of conventional blood culture, making PCR suited to the routine diagnosis of candidaemia [11,12,13]. In fact, PCR lends itself as a good tool for diagnosing candidaemia in high-risk patients. One of the cornerstones of the efficiency of PCR is DNA extraction, which is particularly dependent upon the quality and quantity of the initial material [14]. Thus, choosing an appropriate DNA extraction method is a critical step in the diagnostic laboratory workflow. DNA extraction needs to: (i) be highly efficient for DNA recovery from yeast, which are characterized by a highly complex and solid cell wall [15]; (ii) detect yeasts in low abundance, with a limit of one colony forming unit per milliliter (CFU/mL) [7]; and (iii) remove potential PCR inhibitors.

The extraction/purification of nucleic acids includes two primer steps as follows: cell lysis and separation of nucleic acids from lysate [16]. There are many automated DNA extraction methods available on the market with an efficiency which may be pathogen and/or sample matrix dependent [16,17], but no versatile method has yet been approved. The overall strategy concerning the DNA extraction of fungi is the use of an aggressive prior lysis step among cell lysis techniques including chemical lysis, mechanical lysis, ultrasonic lysis, thermal lysis and enzymatic lysis [16].

Moreover, in recent years, the diagnosis of candidaemia has been increasingly incorporated into a syndromic approach to bloodstream infection diagnosis. Thus, automated DNA extraction methods are pooled in a clinical laboratory and aim to detect a comprehensive array of microorganisms (such as viruses, bacteria and fungi) involved in blood-stream infections. It therefore appears to be essential to evaluate the efficacy of these methods. The main objective of this study was to evaluate the efficacy of eleven automated DNA extraction protocols on human blood specimens artificially spiked with *Candida* yeasts. The secondary objective was to compare the performance of these extraction protocols on the five main species implicated in candidemia (*Candida albicans*, *Candida glabrata*, *Candida parapsilosis*, *Candida tropicalis* and *Candida krusei*).

## 2. Materials and Methods

### 2.1. Preparation of Artificially Inoculated Blood

Blood samples spiked with *Candida* species were prepared as follows. *Candida albicans* ATCC 90,028, *Candida glabrata* (MH545,924), *Candida krusei* ATCC 6258, *Candida parapsilosis* ATCC 22,019 and *Candida tropicalis* (CP047,875) were cultured on Sabouraud dextrose agar plates supplemented with gentamicin and chloramphenicol (Bio-Rad, Marnes-la-Coquette, France) for 48 h at 30 °C. Yeasts cells were suspended in sterile saline solutions (API^TM^ NaCl 0.85% Medium, Biomérieux, Marcy-l’Etoile, France). A rich suspension of blastospores was prepared in a small volume of saline. Fresh EDTA-treated human blood from healthy blood donors (Convention N°7831, “Etablissement Français du Sang”, Marseille, France) was inoculated with this suspension of yeasts cells to obtain eight concentrations of spiked blood samples: 0 CFU/mL, 10 CFU/mL, 50 CFU/mL, 10^2^ CFU/mL, 10^3^ CFU/mL, 10^4^ CFU/mL, 10^6^ CFU/mL and 10^8^ CFU/mL. Infected blood specimens were then aliquoted in 200 µL and stored at −80 °C before DNA extraction.

### 2.2. DNA Extraction Methods

The DNA of each aliquot was extracted in duplicate using the following 11 extraction methods. The characteristics of each DNA extraction method are summarized in Table 1.

Method 1: DNA extraction was performed on 200 µL of whole blood inoculum by using a NucliSENS^TM^ easyMAG^TM^ system (BioMérieux, France) according to the manufacturer’s instructions with a protocol optimized by pre-treatment [18]. In order to achieve maximum yield, a whole blood specimen was pre-treated in a tube containing ceramic beads (Lysing matrix D tube, MP Biomedicals Germany GmbH, Eschwege, Germany) with 500 µL of NucliSENS^TM^ easyMAG^TM^ lysis buffer (BioMérieux, Marcy-l’Etoile, France) and then disrupted in a FastPrep BIO 101 apparatus (Qbiogene, Strasbourg, France) at maximum power for 40 s. The tubes were then centrifuged at 10,000× *g* for one minute. The procedure was then performed on 200 µL of supernatant.

Method 2: DNA extraction was performed directly on 200 µL of a whole blood specimen using the EZ1^TM^ DNA Blood 200 µL Kit (Qiagen, Hilden, Germany) with the 10,591,402 V1.0 DNA blood card in an EZ1 Advanced XL extractor following the manufacturer’s recommendations.

Method 3: DNA extraction was performed on 200 µL of a whole blood specimen by using the EZ1^TM^ DNA Blood 200 µL Kit (Qiagen, Hilden, Germany) with the 10,591,402 V1.0 DNA blood card in an EZ1 Advanced XL extractor supplemented by the pre-treatment procedure used in Method 1.

Method 4: DNA extraction was performed on 190 µL of a whole blood inoculum using the EZ1^TM^ DNA Tissue Kit (Qiagen, France) with the 10,677,990 V1.0 DNA bacteria card in an EZ1 Advanced XL extractor. In order to achieve maximum yield, the pre-treatment consisted of a digestion with 10 µL of Lyticase (25 units/µL, SigmaAldrich, Saint-Louis, MO, USA) at 30 °C for 30 min following the manufacturer’s recommendations.

Method 5: DNA extraction was performed on 200 µL of a whole blood specimen using the QIAamp^TM^ 96 DNA QIAcube HT kit (Qiagen, France) following the manufacturer’s recommendations.

Method 6: DNA extraction was performed on 150 µL of a whole blood specimen using the Macherey-Nagel™ Pathogène NucleoMag™ kit (Fisherscientific, Waltham, MA, USA) in a KingFisher Flex (Thermofisher scientific, Waltham, MA, USA) following the manufacturer’s recommendations.

Method 7: DNA extraction was performed on 200 µL of a whole blood specimen using the Mag-Bind^TM^ Viral DNA/RNA kit (Omega Bio-tek, Norcross, GA, USA) in a KingFisher Flex (Thermofisher Scientific, France) following the manufacturer’s recommendations.

Method 8: DNA extraction was performed on 400 µL of a whole blood specimen using the MagMAX™ Viral/Pathogen Nucleic Acid Isolation Kit (Applied Biosystems, Foster City, CA, USA) in a KingFisher Flex (Thermofisher scientific, France) following the manufacturer’s recommendations.

Method 9: DNA extraction was performed on 200 µL of a whole blood specimen using the Chemagic Viral DNA/RNA 300 kit H96 (PerkinElmer, Waltham, MA, USA) in a Chemagic 360 instrument (PerkinElmer, Waltham, MA, USA) following the manufacturer’s recommendations.

Method 10: DNA extraction was performed on 200 µL of a whole blood specimen using the Virus DNA/RNA Extraction Kit (Wuhan MGI Tech Co., Wuhan, China) in a MGISP-960 instrument (Wuhan MGI Tech Co., Wuhan, China) following the manufacturer’s recommendations.

Method 11: DNA extraction was performed on 200 µL of a whole blood specimen using the Bioextract^TM^ Superball^TM^ kit (Biosellal, Dardilly, France) in a KingFisher Flex (Thermofisher scientific, France) following the manufacturer’s recommendations.

### 2.3. Real Time PCR Assay

Following two extractions with each protocol, all DNA extracts (16 per method in total) were tested in duplicate by PCR. Real-time PCR was carried out using a pan-*Candida* primer set with a *Candida* spp.-specific probe targeting the internal transcribed spacer 2 (ITS2) region of nuclear ribosomal DNA [19]. Primer and probes sequences were provided by Eurogentec (Angers, France).

PCR assays were performed on a LightCycler^TM^ 480 (Roche Diagnostics, Bâle, Switzerland) instrument. Individual real time PCR reactions were carried out in 17 µL of volume in a 96-well plate (Roche Diagnostic) containing 15 µL Master mix (Roche Diagnostics GmbH, Mannheim, Germany), 900 nM of forward primer (CCTGTTTGAGCGTCRTTT), 900 nM of reverse primer (TCCTCCGCTTATTGATAT), 250 nM of specific TaqMAn^TM^ probe [*Candida albicans* (6FAM-TGCTTGCGGCGGTA), *Candida glabrata* (6FAM-TTTACCAACTCGGTGTTGAT), *Candida krusei* (6FAM-GCCGAGCGAACTAGACTTT), *Candida parapsilosis* (6FAM-GAAAGGCGGAGTATAAAC) or *Candida tropicalis* (6FAM-GGCCACCACAATTTATTTCA)] and 2 µL of DNA. No-template PCR controls were included in each run. The thermal cycling conditions were 95 °C for 10 min, followed by 45 cycles of 95 °C for 10 s, 54 °C for 30 s and 72 °C for 10 s.

### 2.4. Detection of Inhibitors

To detect PCR inhibitors, all DNA samples were tested both pure and after a 20-fold dilution. The expected difference in Ct values between the two concentrations is 4.33 in the absence of inhibitors. Partial inhibition was defined as a less than 3.5 difference in Ct values [18].

### 2.5. DNA Quantification

Extracted DNA was quantified using the NanoDrop^TM^ ND-1000 (Thermo Scientific, Waltham, MA, USA) according to the manufacturer’s instructions.

### 2.6. Human Gene Amplification

The human albumin gene, as a DNA extraction control, was amplified in each DNA extract on a LightCycler^TM^ 480 (Roche Diagnostics, France) instrument. Individual real time PCR reactions were carried out in 17 µL volume in a 96-well plate (Roche Diagnostic) containing 15 µL Master mix (Roche Diagnostics GmbH, Mannheim, Germany), 500 nM of forward primer (GCTGTCATCTCTTGTGGGCTGT), 500 nM of reverse primer (AAACTCATGGGAGCTGCTGGTTC), 250 nM of specific TaqMAn^TM^ probe (6FAM-CTGTCATGCCCACACAAATCTCTCC). The thermal cycling conditions were 95 °C for 10 min, followed by 45 cycles of 95 °C for 10 s and 60 °C for 30 s.

### 2.7. Specimen Testing

For each *Candida* species, DNA of the eight blood inocula (i.e., 0, 10, 50, 10^2^, 10^3^, 10^4^, 10^6^ and 10^8^ CFU/mL) was extracted in duplicate. Real time PCR for each of the 16 specimens was carried out in duplicate. PCR results were considered negative when the cycle threshold (Ct) value exceeded 45 or when no amplification curve was obtained.

When a single test for the four samples was positive, the tests were again, in order to eliminate cross contamination. A positive result was defined as at least two positive results among eight replicates of the same inoculum.

### 2.8. Determination of PCR Sensitivity

The efficiency of the *Candida* specific PCR was evaluated by plotting a standard curve with a serial 10-fold dilution (1 to 10^10^ copy number/µL) of plasmid DNA (20AD2FVC_Candida_PMA-RQ). The efficiency, slope and Y intercept were calculated with the LightCycler^TM^ 480 Real-Time PCR System software.

### 2.9. Statistical Analysis

Statistical analysis was performed using STATA 14.0 and R version 4.0 (package {ggplot2}) [20] was used for additional graphical representation.

Negative inocula were annotated at 45 Ct, either the maximum number of PCR cycles performed. Mean Ct value and 95% confidence intervals (95% CI) were calculated for each DNA extraction method. Comparisons of the mean Ct values of human albumin amplification obtained with each method used the Kruskal–Wallis non-parametric test. The crude detection rate was defined as the number of positive *Candida* PCRs over the total number of reactions performed for a given method. The 95% CI were calculated, and detection rates were compared across methods using the Chi-square test.

To compare methods, the primary outcome was the detection of the pathogen by qPCR following extraction and coded as a binary variable (detected/non detected). The effects of the extraction method, the pathogen species and the concentration of the spiked sample were analyzed using multivariate multilevel logistical regression. In order to account for the correlation between the results for duplicate PCR testing of the DNA products obtained from a single extraction; a random effect was included at the extraction level. An interaction between the pathogen and method was tested using the likelihood ratio test (model with interaction nested in model without). Sample concentrations were included as an independent covariable. For each pathogen, we ranked the methods based on their detection performance in relation to the method with the best overall detection using the adjusted odds-ratios, 95% confidence intervals (95%CI) and *p*-values adjusted for sample concentration obtained from the multivariate model. Extraction methods were considered as performing significantly worse than the reference if the upper limit of their 95% CI was below one. Apart from the reference, two methods were considered to be significantly different from one another if they had non-overlapping 95% CI.

## 3. Results

### 3.1. Determination of PCR Performances

The results of efficiency, slope and Y intercept for each species-specific PCR are as follows: *Candida albicans* (135%; −2.315; 29.51), *Candida glabrata* (102%; −3.230; 37.44), *Candida krusei* (98.7%; −3.386; 39.08), *Candida parapsilosis* (126%; −2.484; 30.84) and *Candida tropicalis* (111.7%; −2.864; 34.64).

### 3.2. Human Albumin Gene Amplification and DNA Quantification

The human albumin gene was amplified to constitute a complete process control for DNA extraction. The results were expressed as the mean of Ct values obtained for all DNA extracts from a given extraction method (Figure 1A). Considering an extraction method, the Ct values of albumin-PCR do not differ according to *Candida* species or the inocula tested. A significant difference was observed between the 11 DNA extraction methods (*p* ≤ 0.001) (Kruskal–Wallis test). We observed three groups: Methods 1–5 with the lowest average Ct and narrow 95%CI, Methods 6–7 with the highest average Ct and narrow 95%CI, and Methods 8–11 with high Ct values and broad confidence intervals. Overall, Methods 1–5 performed significantly better than 6–7, and Methods 8–11 had intermediate performance. Using Nanodrop^TM^ to quantify the extracted DNA, a heterogeneity between the extraction methods was found. No correlation was observed between the Ct of albumin and the amount of template DNA in the sample. Thus, Method 7 gave the highest quantity of DNA yield (84.5 ± 29.0) while it is one of the methods with highest average Ct.

### 3.3. Comparison of DNA Extraction Methods in Isolating Candida DNA from Spiked Blood

All negative control whole blood samples (*n* = 55) remained negative. The global performances of the automated DNA extraction of a range of yeast concentrations were expressed as an overall detection rate (Figure 1B). Only Method 1 yielded more than 80% positive results. Method 3 yielded more than 60% positive results. Methods 4 and 8 showed 59.0% and 54.7% positive results respectively. All the other methods had a detection rate below 50%, significantly lower than Methods 1, 3, 4 and 8 with positive results lower to 50%. Methods 5–7, 9 and 11 in particular presented detection rates of below 35%. Method 1 showed a homogenous distribution of Ct ranging from 15.8 to 40, rising with the gradient of the inoculum (Figure 2). Ct levels less than 20 were obtained only after DNA extraction with Methods 1, 3 and 4. Taking into account the positivity rate and the distribution of Ct, Method 1 appeared to be the most efficient, for all species and for all inocula and was therefore chosen as reference method for the next statistical analysis. Method 3, corresponding to Method 2 optimized with a mechanical pre-treatment protocol, also appears to yield acceptable results.

### 3.4. Performance of DNA Extraction Methods Adjusted for Candida Species and Sample Concentration

All samples were positive for a concentration of 10^8^ CFU/mL and this concentration was excluded in the subsequent analysis. We observed a significant difference (Chi-square test, *p* ≤ 0.001) in DNA extraction efficiency between the five species tested: *Candida albicans*, *Candida glabrata* complex, *Candida parapsilosis*, *Candida tropicalis* and *Candida krusei* (Figure 3).

The performances of automated protocols for DNA extraction for each *Candida* species were expressed as the odds ratio of PCR detection, adjusted for inoculum concentration using a multilevel logistical regression model. Method 1 was used as a reference method to make it easier to rank the other methods. A statistically significant interaction was identified between *Candida* species and method (*p* < 0.0001, likelihood ratio test), indicating that methods could perform differently for a given species, and results from the model with interaction are presented.

Only Method 1 was effective for *Candida tropicalis* DNA extraction, with a probability of detection >50%, even for the low amount of blastospores (i.e., 10 CFU/mL). Its performance for *C. parapsilosis* also appeared higher than in Methods 3 and 4, without reaching statistical significance. With regards to *C. albicans*, *C. glabrata* and *C. krusei* no significant difference between the reference Method 1 and Methods 3 and 4 in terms of DNA extraction efficiency was observed (Appendix A). *Candida krusei* appears to be less susceptible to extraction methods. After adjusting for blastospore concentration, no significant differences were observed in the detection of *C. krusei* between the reference Method 1 and Methods 3, 4, 8 and 10. Method 8 showed irregular efficiency for *C. albicans* and *C. glabrata* DNA extraction.

### 3.5. Detection of PCR Inhibitors

No inhibitors were detected in any DNA extract. All of the eleven extraction methods apparently performed equally well in eliminating PCR inhibitors.

## 4. Discussion

In this study, we evaluated the suitability of 11 recent automated procedures for the isolation of *Candida* DNA from artificially spiked blood samples: NucliSENS^TM^ EasyMAG^TM^ (BioMérieux) (Method 1), EZ1^TM^ DNA Blood 200 µL Kit (Qiagen) (Method 2), EZ1^TM^ DNA Blood 200 µL Kit with pre-treatment (Qiagen) (Method 3), EZ1^TM^ DNA Tissue Kit with pre-treatment (Qiagen) (Method 4), QIAamp^TM^ 96 DNA QIAcube HT Kit (Qiagen) (Method 5), Macherey-Nagel™ Pathogène NucleoMag™ (Fisher Scientific) (Method 6), Mag-Bind^TM^ Viral DNA/RNA (Omega Biotek) (Method 7), MagMAX™ Viral/Pathogen Nucleic Acid Isolation Kit (Applied Biosystems) (Method 8), Chemagic Viral DNA/RNA 300 kit H96 (PerkinElmer) (Method 9), Virus DNA/RNA Extraction kit (MGI) (Method 10) and Bioextract^TM^ Superball^TM^ (Biosellal) (Method 11). These extraction methods were tested over a wide sequential range of blastospore concentrations from 10 CFU/mL to 10^8^ CFU/mL. The detection of *Candida* DNA in spiked blood was performed by an *in-house* PCR targeting the ITS2 region of nuclear ribosomal DNA with an efficiency greater than 98% [19]. It should be noted that there are many in-house or commercially available real-time PCR assay kits targeting various genetic sequences (18S rDNA, 28S rDNA, 5.8S rDNA, ITS regions and mitochondrial DNA) [12,13,21,22]. All DNA extracts were tested in duplicate, which gave reproducible results in the majority of cases, for all methods and all concentrations, although there were gaps for high (>37) cycle thresholds.

As no consensus has been found concerning the best blood fraction to be tested for the diagnosis of candidaemia [23], like Metwalli et al. [24], we reasoned that inoculating fresh uninfected EDTA-treated human blood with *Candida* species, would best mimic the real conditions of candidaemia. This implies that the methods are suitable for extracting DNA from media that are rich in cells and PCR inhibitors [25]. The difference before and after 20-fold dilution demonstrates that all 11 methods were able to remove PCR inhibitors. Surprisingly, no correlation was found between the Ct values and the amount of template DNA in the sample. Concerning the Chemagic Viral DNA/RNA 300 kit H96 (PerkinElmer), probably the red tinted sample disrupted DNA quantification. Few studies comparing extraction methods quantify DNA, suggesting that house-keeping gene amplification is a better of evaluating extraction efficiency. However, we observed that the extraction methods were not equally efficient in isolating human DNA. We thus distinguished two main groups, the best results being obtained with the four Qiagen automated procedures namely: EZ1^TM^ DNA Blood 200 µL Kit (Qiagen), EZ1^TM^ DNA Blood 200 µL Kit with pre-treatment (Qiagen), EZ1^TM^ DNA Tissue Kit with pre-treatment (Qiagen), QIAamp^TM^ 96 DNA QIAcube HT Kit (Qiagen) and NucliSENS^TM^ EasyMAG^TM^ (BioMérieux). With the exception of the Chemagic Viral DNA/RNA 300 kit H96 (PerkinElmer) and Virus DNA/RNA Extraction Kit (MGI), all the extraction methods tested were designed for use on whole blood. The Bioextract^TM^ Superball^TM^ (Biosellal), is provided for veterinary laboratories and exhibits higher Ct values with respect to human albumin gene amplification.

Few studies have compared current automated nucleic acid extraction methods for the isolation of DNA of the five main *Candida* species from whole blood [24,26,27,28]. Here, all 11 extraction methods were equivalent at concentrations greater than 10^6^ CFU/mL. At the lowest concentration (between 10 CFU/mL and 100 CFU/mL), the NucliSENS^TM^ easyMAG^TM^ (BioMérieux) procedure stood out significantly from the ten other methods. We specifically tested low blastospore concentration, as these concentrations are relevant in a clinical setting. The NucliSENS^TM^ easyMAG^TM^ system had previously been optimized to allow for the extraction of fungal DNA [16]. Interestingly, no significant difference was observed between the EZ1^TM^ DNA Blood 200 µL Kit with pre-treatment (Qiagen) and the EZ1^TM^ DNA Tissue Kit with pre-treatment (Qiagen) with regard to the detection rate (66.7% and 59.0% respectively). These two methods were optimized by the addition of a pre-treatment comprising chemical and mechanical lysis for one, and enzymatic lysis for the other. The extraction procedure EZ1^TM^ DNA Blood 200 µL kit was evaluated with and without pre-treatment. Optimization of the protocol by adding a chemical and mechanical pre-treatment have be relevant with a significant improvement in the *Candida* detection rate, especially for concentrations lower than 10^4^ CFU/mL (40.6% without pre-treatment vs. 66.7% with pre-treatment). This is in line with a previous study that clearly demonstrated that introducing a bead beating step to the EZ1 procedure improved fungal DNA extraction from human specimens [28]. *Candida* species have particularities concerning their yeast cell wall, which must have an impact on their lysis susceptibility. The QIAamp^TM^ 96 DNA QIAcube HT Kit (Qiagen), Macherey-Nagel™ Pathogène NucleoMag™ (Fisher Scientific), Mag-Bind^TM^ Viral DNA/RNA (Omega Biotek), Chemagic Viral DNA/RNA 300 kit H96 (PerkinElmer) and Bioextract^TM^ Superball^TM^ (Biosellal) resulted in the worst efficiencies, with less than 35% of the sample being detected. It should be noted that some of them were validated for viral DNA. Interestingly, the performances of extraction methods when amplifying the albumin gene mirror those performance in *Candida* PCR. This paper highlights a difference in DNA extraction efficiency between the five different species mainly involved in invasive candidiasis. For *C. albicans*, *C. glabrata* and *C. krusei* DNA extraction, three methods, namely NucliSENS^TM^ easyMAG^TM^, EZ1^TM^ DNA Blood 200 µL Kit with pre-treatment and EZ1^TM^ DNA Tissue Kit with pre-treatment, are equivalent and effective. It is notable that, for the species *C. tropicalis*, only the NucliSENS^TM^ easyMAG^TM^ procedure has been shown to be effective with a probability of detection >50%, even for the low amount of blastospore (i.e., 10 CFU/mL). The difference in efficiency of the DNA extraction methods according to *Candida* species may be explained by the difference in matrix and composition of their wall in terms of filamentation capacity, the quantity of matrix carbohydrates, protein, and also its cell-surface hydrophobicity [29]. These characteristics, specific to each species, have been evaluated in their biofilm-forming capacity. Thus, *Candida tropicalis* has shown a higher biofilm-forming capacity than *C. krusei*, *C. parapsilosis* and *C. albicans*, characterized by high hydrophobicity, and its ability to form a very dense and intertwined biofilm [29], which may explain the need for more aggressive DNA extraction. The difference between species can also be related to a heterogeneous compatibility between certain extraction methods and a given PCR species assay. Despite these differences, NucliSENS^TM^ easyMAG^TM^ yielded the best results for the five *Candida* species.

Finally, in the era of the syndromic approach to bloodstream infection diagnosis, the purification of DNA from various organisms must be performed simultaneously using this type of extraction system with just a single extraction method. It is, therefore, necessary to evaluate these automated pieces of equipment for each pathogen (fungus, bacteria and virus). Of the procedures tested in this study, seven allowed for the simultaneous extraction of both DNA and RNA, including the NucliSENS^TM^ easyMAG^TM^ procedure. This versatile technique thus stands out from the other techniques tested here, allowing for the efficient isolation of DNA from the five species of *Candida* involved in human pathology.

## 5. Conclusions

The present study aimed to evaluate eleven automated DNA extraction protocols on artificially spiked blood specimens with the five main *Candida* yeasts for the suitable routine diagnosis of invasive candidaemia. This is the first study to demonstrate a difference in DNA extraction performance between *Candida* species (*Candida albicans, Candida glabrata* complex, *Candida parapsilosis*, *Candida tropicalis* and *Candida krusei*). Fortunately, one extraction method (i.e., NucliSENS^TM^ easyMAG^TM^ (BioMérieux)) displayed adequate performance for detecting the DNA of five *Candida* species in whole blood samples, which is mandatory for the current syndromic diagnosis of bloodstream infections.

## Figures and Tables

**Figure 1 jof-07-00228-f001:**
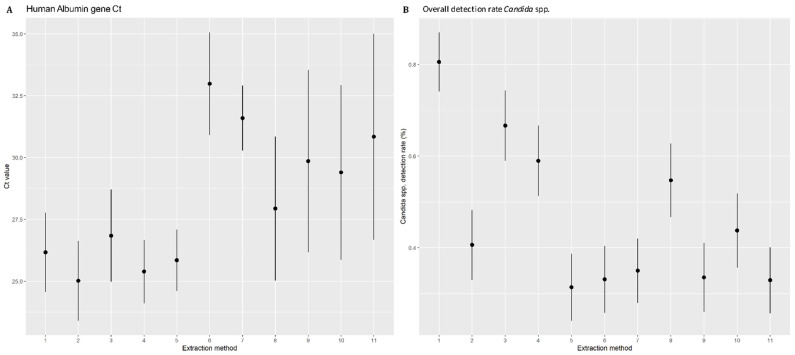
(**A**) Distribution of human albumin gene Ct according to the extraction methods tested. The means of the Cycle threshold and their 95% confidence interval are shown. (**B**) Distribution of the *Candida* species detection rate (in percentages) according to the extraction methods tested. The detection rates (in percentages) and their 95% confidence interval are shown.

**Figure 2 jof-07-00228-f002:**
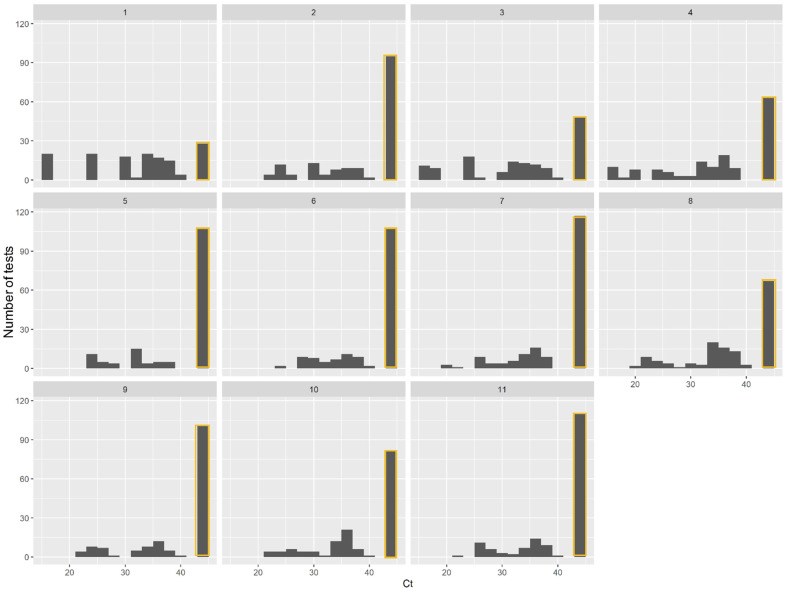
Number of positive *Candida* PCR test per Ct according to the 11 extraction methods. Negative results (Ct > 45 or no amplification detected) are stained orange.

**Figure 3 jof-07-00228-f003:**
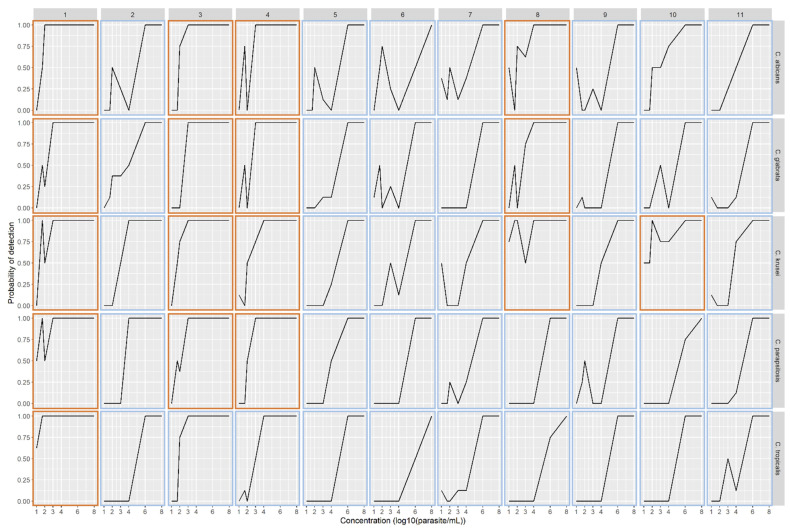
The probability of obtaining at least one PCR positive result (Ct < 45) on the duplicate from the two extractions for each inoculum concentration according to the *Candida* species and the automated method used.

**Table 1 jof-07-00228-t001:** Characteristics of the DNA extraction methods. The DNA extract measured by Nanodrop is expressed as a mean concentration (±standard deviation) C: chemical; M: mechanical; E: enzymatic; LB: NucliSENS^TM^ easyMAG^TM^ lysis buffer (BioMérieux, Marcy-l’Etoile, France); CB; ceramic beads; Ly: lyticase; PK: proteinase K; BALF: broncho-alveolar lavage fluid, DNA: deoxyribonucleic acid; RNA: ribonucleic acid.

Method	Kit	Company	Automate	Validated Sample	Validated Material	Nucleic Acids	Pretreatment			Sample Volume	Expected Elution Volume	Observed Elution Volume	Elution Appearance	DNA Quantification (ng/µL)
							C	M	E					
**1**	NucliSENS^TM^ EasyMAG^TM^	BioMérieux	NucliSENS EasyMAG	Human	Whole blood, serum, plasma, stools, respiratory samples and other body fluids.	DNA/RNA	LB	CB	-	200 µL	110 µL	110 µL	Clear	27.0 (±6.6)
**2**	EZ1^TM^ DNA Blood 200 µL Kit	Qiagen	EZ1	Human	Whole blood	DNA	-	-	-	200 µL	100 µL	100 µL	Clear	30.5 (±5.5)
**3**	EZ1^TM^ DNA Blood 200 µL Kit + pretreatment	Qiagen	EZ1	Human	Whole blood	DNA	LB	CB	-	200 µL	100 µL	100 µL	Clear	10.4 (±2.7)
**4**	EZ1^TM^ DNA Tissue Kit + pretreatment	Qiagen	EZ1	Human	Whole dried blood, tissue, buccal cells, cultured cells, Paraffin-Embedded Tissue	DNA	-	-	Ly	200 µL	100 µL	100 µL	Clear	30.3 (±3.3)
**5**	QIAamp 96 DNA QIAcube HT Kit	Qiagen	QIAcube	Human	Whole blood, tissue, cells	DNA	-	-	PK	200 µL	120 µL	120 µL	Red tinted	30.2 (±6.6)
**6**	Macherey-Nagel™ Pathogène NucleoMag™	Fisher Scientific	KingFisher	Human	Whole blood, serum, plasma; tissue (e.g., ear notches); feces; swab wash solution	DNA/RNA	-	-	PK	150 µL	80 µL	80 µL	Clear	30.2 (±7.9)
**7**	Mag-Bind^TM^ Viral DNA/RNA	Omega Bio-tek	KingFisher	Human	Whole blood, serum, plasma, saliva, and other body fluids.	DNA/RNA	-	-	PK	200 µL	100 µL	100 µL	Clear	84.5 (±29.0)
**8**	MagMAX™ Viral/Pathogen Nucleic Acid Isolation Kit	Applied Biosystems	MGISP-960	Human	Whole blood, swabs, urine, and viral transport media	DNA/RNA	-	-	PK	400 µL	100 µL	100 µL	Red tinted	32.2 (±35.3)
**9**	Chemagic Viral DNA/RNA 300 kit H96	PerkinElmer	Chemagic 360	Human	Serum, plasma, saliva, nasal or oral swab, BALF	DNA/RNA	-	-	PK	300 µL	100 µL	100 µL	Red tinted	17.8 (±19.5)
**10**	Virus DNA/RNA Extraction kit	MGI	MGISP-960	Human	Serum, plasma, saliva, virus culture medium, throat swabs, BALF, sputum	DNA/RNA	-	-	PK	200 µL	100 µL	25 µL	Clear	61.3 (±42.6)
**11**	Bioextract^TM^ Superball^TM^	Biosellal	KingFisher	Veterinary	Whole blood, milk, serum, organs	DNA/RNA	-	-	PK	200 µL	100 µL	100 µL	Clear	13.9 (±2.8)

## Data Availability

Not applicable.

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
