# Peer review of "Evaluation of 11 DNA Automated Extraction Protocols for the Detection of the 5 Mains Candida Species from Artificially Spiked Blood"

_jof, 2021, doi:10.3390/jof7030228_

Round 1

Reviewer 1 Report

In this manuscript the authors compare 11 different DNA extraction methods to detect Candida species in bloodstream infections. This is an essential and useful study to develop standards for detection of Candida in clinical laboratories. However, there are few concerns which need to be addressed.

Introduction: A brief paragraph describing the current methodologies used for extraction of DNA in clinical laboratories in different parts of the world should be included along with their drawbacks. This will emphasize the significance of this study.

Introduction, Line 50 states that a good extraction protocol should detect Candida at as low as 1 CFU/mL. Then, why was not this concentration tested (see lines 73-74).

This reviewer appreciates that thoughtfulness of the authors to test human albumin gene as an extraction control. This is necessary to prove that DNA was extracted at detectable amounts.

That said, one big question is - what is the concentration of DNA extracted by each method? Was it quantified? If so, it should be included in table 1. It is mentioned that 2uL of DNA was used for qPCR. As DNA concentration can directly impact the limit of detection, the authors should discuss the correlation between DNA concentration and observed Ct for each method. This should be included in both results and discussion.

Minor:

There are several typographical and grammatical errors (only some of which are listed below) should be corrected throughout the manuscript.

  1. In entire manuscript "Candida" should be italicized.
  2. Description of methods: Change 'constructor' to 'manufacturer'
  3. Introduction, Line 53: Should be spelled as "Thus"
  4. Methods, Lines 10 and 30: Change 'direct primer' to 'forward primer'
  5. Methods, Lines 41-42: remove these lines. Its duplicated.
  6. Results, Lines 75-77: This sentence (This section may.............can be drawn) should be removed.
  7. Discussion: Line 154; Remove the word "Authors"
  8. Discussion, Line 187: Remove 'use'

Author Response

Responses to the reviewers:

Reviewer 1

In this manuscript the authors compare 11 different DNA extraction methods to detect Candida species in bloodstream infections. This is an essential and useful study to develop standards for detection of Candida in clinical laboratories. However, there are few concerns which need to be addressed.

R: We thank Reviewer 1 for their comments.

Introduction: A brief paragraph describing the current methodologies used for extraction of DNA in clinical laboratories in different parts of the world should be included along with their drawbacks. This will emphasize the significance of this study.

R: As requested by Reviewer 1, a brief paragraph describing the current methodologies used for DNA extraction in clinical laboratories has been included in the introduction section (line 52-58).

Introduction, Line 50 states that a good extraction protocol should detect Candida at as low as 1 CFU/mL. Then, why was not this concentration tested (see lines 73-74).

R: Technically, it is difficult to test a concentration of 1 CFU/mL. We worked on 200 µL of artificially spiked blood samples: a concentration at 1 CFU/mL is equivalent to obtaining only one yeast every five test samples.

In addition, as described in our study, cascade dilutions quickly lead to negative results.

This reviewer appreciates that thoughtfulness of the authors to test human albumin gene as an extraction control. This is necessary to prove that DNA was extracted at detectable amounts.

That said, one big question is - what is the concentration of DNA extracted by each method? Was it quantified? If so, it should be included in table 1. It is mentioned that 2uL of DNA was used for qPCR. As DNA concentration can directly impact the limit of detection, the authors should discuss the correlation between DNA concentration and observed Ct for each method. This should be included in both results and discussion.

R: As requested by the reviewer, we have used Nanodrop technology to quantify the concentration of the DNA extracted by each method and have integrated these results in Table 1. We showed heterogeneity between the extraction methods. No correlation was observed between the Ct of albumin and the amount of template DNA in the sample. This has been included in the results section (line 23) and the discussion section “Surprisingly, no correlation was found between the Ct values and the amount of template DNA in the sample. Concerning the Chemagic Viral DNA/RNA 300 kit H96 (PerkinElmer), probably the red tinted sample disrupted DNA quantification.”

Minor:

There are several typographical and grammatical errors (only some of which are listed below) should be corrected throughout the manuscript.

1.In entire manuscript "Candida" should be italicized.

R: We agree with the reviewer. Scientific names of species were italicized accordingly.

2.Description of methods: Change 'constructor' to 'manufacturer'

R: We agree with the reviewer. We have changed ‘constructor’ to ‘manufacturer’.

3.Introduction, Line 53: Should be spelled as "Thus"

R: We agree with the reviewer and have corrected the mistake.

4.Methods, Lines 10 and 30: Change 'direct primer' to 'forward primer'

R: We agree with the reviewer. We have changed 'direct primer' to 'forward primer'.

5.Methods, Lines 41-42: remove these lines. Its duplicated.

R: We agree with the reviewer. We have removed these lines.

6.Results, Lines 75-77: This sentence (This section may.............can be drawn) should be removed.

R: We agree with the reviewer. We have removed this sentence.

7.Discussion: Line 154; Remove the word "Authors"

R: We agree with the reviewer. We have removed the word "Authors".

8.Discussion, Line 187: Remove 'use'

R: We agree with the reviewer. We have removed the word “use”.

Reviewer 2 Report

Evaluation of 11 DNA automated extraction protocols for the detection of the 5 mains Candida species from artificially spiked blood

Menu et al. investigated the suitability of 11 recent automated procedures for the extraction of low and high amounts of Candida DNA from spiked blood. They found NucliSENSTM easyMAGTM performing the best among the tested protocols. Additionally, they found that the extraction efficiency ranging between 31.4% to 80.6%.

Introduction:

The introduction section describes the main parts of the research, but it suffered from English grammar and vocabulary mistakes it makes it hard to follow. 

Methods

Line 51 ggplot citation: 

Wickham H (2016). ggplot2: Elegant Graphics for Data Analysis. Springer-Verlag New York. ISBN 978-3-319-24277-4,

Author Response

Responses to the reviewers:

Review 2

Evaluation of 11 DNA automated extraction protocols for the detection of the 5 mains Candida species from artificially spiked blood

Menu et al. investigated the suitability of 11 recent automated procedures for the extraction of low and high amounts of Candida DNA from spiked blood. They found NucliSENSTM easyMAGTM performing the best among the tested protocols. Additionally, they found that the extraction efficiency ranging between 31.4% to 80.6%.

R: We thank Reviewer 2 for their comments.

Introduction:

The introduction section describes the main parts of the research, but it suffered from English grammar and vocabulary mistakes it makes it hard to follow.

R: the manuscript has now been edited by a native English speaker and professional proofreader.

Methods

Line 51 ggplot citation:

Wickham H (2016). ggplot2: Elegant Graphics for Data Analysis. Springer-Verlag New York. ISBN 978-3-319-24277-4,

R: We agree with Reviewer 2 and have added this reference to the Methods section.

Round 2

Reviewer 1 Report

All comments were addressed.